# An exponential build-up in seismic energy suggests a months-long nucleation of slow slip in Cascadia

Claudia Hulbert [1,2 ✉], Bertrand Rouet-Leduc [2], Romain Jolivet [1,3] & Paul A. Johnson[2]

Slow slip events result from the spontaneous weakening of the subduction megathrust and bear strong resemblance to earthquakes, only slower. This resemblance allows us to study fundamental aspects of nucleation that remain elusive for classic, fast earthquakes. We rely on machine learning algorithms to infer slow slip timing from statistics of seismic waveforms. We find that patterns in seismic power follow the 14-month slow slip cycle in Cascadia, arguing in favor of the predictability of slow slip rupture. Here, we show that seismic power exponentially increases as the slowly slipping portion of the subduction zone approaches failure, a behavior that shares a striking similarity with the increase in acoustic power observed prior to laboratory slow slip events. Our results suggest that the nucleation phase of Cascadia slow slip events may last from several weeks up to several months.

[1] Laboratoire de Géologie, Département de Géosciences, École Normale Supérieure, PSL Université, CNRS UMR 8538, Paris, France. [2] Los Alamos National Laboratory, Geophysics Group, Los Alamos, NM, USA. [3] Institut Universitaire de France, 1 rue Descartes, 75005 Paris, France. ✉email: claudia.hulbert@ens.fr

Since their discovery in Japan at the turn of the millennium[1,2], slow slip events and associated tectonic tremor and low-frequency earthquakes (LFEs) have been identified in most subduction zones as well as other tectonic environments[2–9]. Slow slip events release energy over much longer durations than classic earthquakes, from a few days to months or even years[2]. In subduction settings, slow slip occurs deep along the subduction interface (i.e. roughly 50 km depth), down-dip from the nucleation zone of damaging earthquakes, at the transition from brittle to ductile deformation[10]. At such depths, slow slip and tremor are thought to take place where temperatures drive dehydration of subducting material that increases pore pressure, inhibiting dynamic failure in the brittle/ductile transition regime[11,12]. The slowly slipping region is considered to mark the transition from unstable (seismogenic) to stable (creeping) sliding and therefore may define the depth limit of megathrust ruptures[12].

A growing body of literature suggests that slow, aseismic slip and rapid, seismic slip bear strong resemblance[9,13,14]. In particular, recent studies find that they follow comparable scaling relationships in terms of duration and magnitude[14–16]. Slow slip events may therefore provide an opportunity to study fundamental rupture physics, as they take place over long periods of time without radiating large amplitude seismic waves. Considering the lack of observational evidence of earthquake nucleation mechanisms, we propose to explore the period leading up to a slow slip event as a window into a better understanding of the nucleation of a slip instability in nature. Here, we transpose a methodology developed on laboratory experiments to the occurrence of slow slip events in Cascadia.

Laboratory studies of slow slip[13] from a bi-axial shear device[17–19] suggest that the amplitude of acoustic noise coming from a fault follows characteristic patterns throughout the slip cycle. Such patterns allow us to estimate key properties of the laboratory fault, including friction on the fault as well as fault displacement rate. In a first effort to generalize these results to a natural fault system, the analysis of slow slip in Cascadia[20] revealed that statistical characteristics of continuous seismic signals can be used to estimate the displacement rate of GPS stations at the surface. These characteristics are related to seismic power, which is analogous to the acoustic power measured in laboratory experiments (we define seismic power as the average of seismic energy per unit of time, i.e. the squared measured ground velocity per unit of time).This proportionality between seismic power and surface displacement enables a quantitative characterization of slow slip events from seismic data.

Machine learning (ML) analysis of seismic data is an expanding field, with recent studies focusing on event detection[21], phase identification[22], phase association[23,24], or patterns in seismicity[25]. In the following, we investigate whether seismic signatures can be found in the period leading up to any known manifestation of major slow slip occurrence anywhere in the Cascadia region. We find that for most episodic slip and tremor events, features within tectonic tremor frequency bands increase a few months before any detection of cataloged tremor or any geodetic signature of fault slip is made in the Cascades. We interpret this growth in the seismic power of the subduction zone as the signature of a nucleation phase that can be detected long before being observed in tremor catalogs or GPS data.

## Results

### Seismic power analysis and the occurrence of slow slip in Cascadia.

We analyze seismic data on Vancouver Island, Canada, where the Juan de Fuca oceanic plate subducts beneath the North American plate (Fig. 1a). The quasi-periodic occurrence of slow slip events (approximately every 14 months)[6,26–30] is manifested by the North American plate lurching southwesterly over the Juan de Fuca plate, generating bursts of tectonic tremor over the area (Fig. 1b). Smaller slow slip events occurring between these large periodic events have been identified recently, pointing towards a large variability in the size and timing of slow earthquakes in the area[31]. The regular occurrence of slow slip events in Cascadia, especially in the vicinity of Vancouver Island[26], compares well to the afore-mentioned laboratory experiments. Furthermore, continuous seismic recordings in this region are available for well over a decade. Supervised ML used in the work described below requires robust training and testing sets including many slip events. The long history of recurrent slow slip observed in this region makes it an ideal case to (i) apply a methodology that has been developed in the laboratory and (ii) determine if there is information carried in the seismic signal characteristic of nucleation and upcoming failure.

We rely on the Pacific Northwest Seismic Network (PNSN) Tremor Logs[29] to identify slow slip failures. The Tremor Logs report ongoing slow slip on the basis of tremor activity detected anywhere in Cascadia by the PNSN. In what follows, the Tremor Logs will be used to identify slow slip timings. These timings are represented by vertical gray bars in Fig. 2b. Smoothed tremor rates in Vancouver Island from the PNSN tremor catalog[29] are also plotted for comparison.

Tremor logs take into account the entire Cascadia region and are not geographically limited to Vancouver Island. This is important for our analysis, as it precludes contamination of seismic data by an ongoing slow slip event. Indeed, slow slip events in Cascadia take place over a large section of the west coast of the USA and Canada, and the beginning and migration of events do not follow systematic patterns. Preceding any local manifestation of slow slip activity, seismic data might include information from an event that has already started elsewhere and is migrating towards our region of interest.

For this reason, using GPS displacement or only local tremor to identify failure times would not be robust for our analysis. GPS displacement is a local measurement, and so is the occurrence of local tremor, whereas seismic sensors may capture signatures of slipping segments located farther away. Furthermore, identifying slow slip with GPS requires temporal smoothing and consequently does not allow one to determine precisely when the rupture begins, which may introduce errors of days to weeks and ultimately arbitrarily improve the performance of our analysis. This explains why we favor the PNSN tremor logs to determine slip timing (note that tests using local peaks in tremor rates or local GPS displacement as a proxy for failure times provides similar or better results—see Supplementary).

We rely on nine seismic stations from the Plate Boundary Observatory[32] (Fig. 1b). We find that borehole stations are much more robust than surface ones because of contamination by seasonal signals. A continuous, clipped and de-sampled seismic waveform for one seismic station is shown in Fig. 2a. We first process the seismic data by correcting for the instrument gain, for each day during the period analyzed (2005–2018). The daily data intervals are band-passed between 8 and 13 Hz (within successive bands of 1 Hz) and clipped, to limit the contribution of microseismic noise and earthquakes, respectively, and to focus on low-amplitude signals. These particular frequency bands have been identified in our previous work as the most informative of the behavior of the system[20]. Once the data are pre-processed, for each day we compute a number of statistical features linked to signal energy. Building on our previous work, the features correspond to inter-quantile ranges of seismic data within tremor frequency bands (8–13 Hz, by 1 Hz increments). These features are representative of seismic energy in tremor frequency bands,

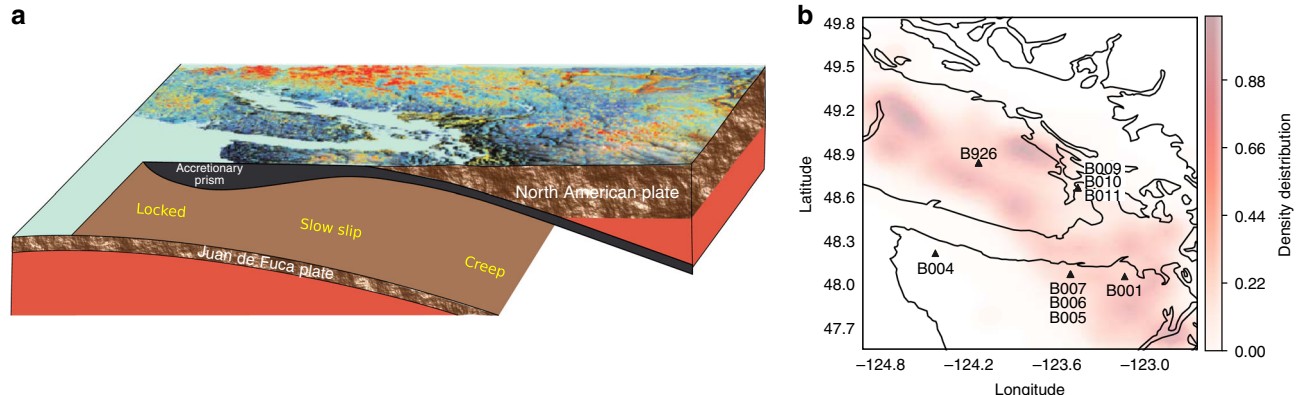

**Fig. 1 Schematic of the subduction zone and seismic array analyzed. a** Sketch of the subduction zone underneath Vancouver Island, Canada. **b** Map of the region analyzed. The seismic stations used in this study are shown with black triangles. Red shaded areas correspond to the density distribution of tremors from the PNSN tremor catalog from 2009 to late 2018.

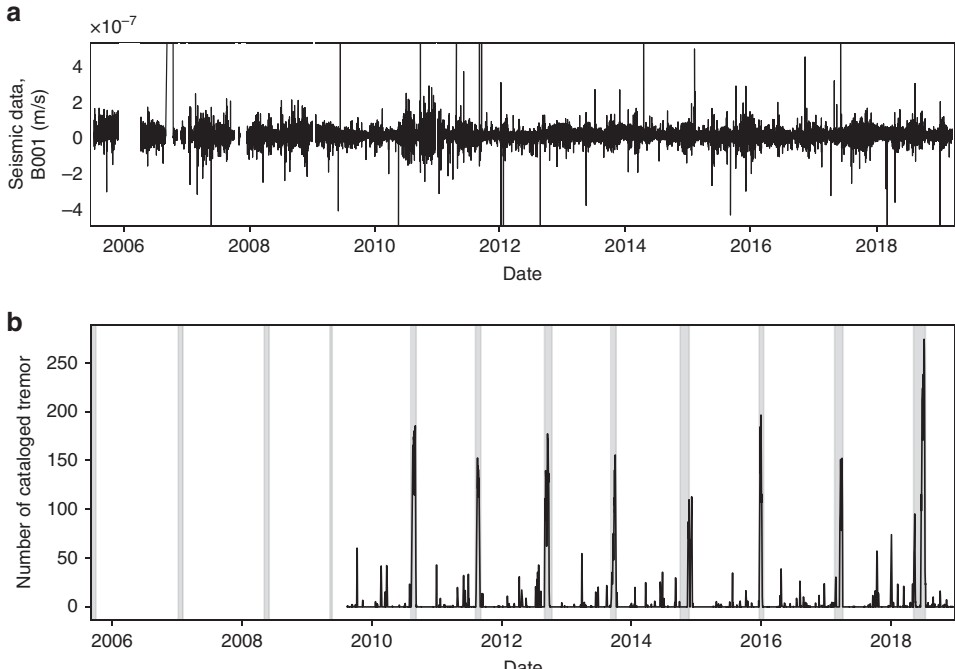

**Fig. 2 Signals analyzed and timings of slow slip events. a** Down-sampled (solely for visualization purposes), clipped (at 5E−7 m/s) continuous seismic waves for one station analyzed (B001). Our goal is to rely exclusively on continuous seismic waves to identify signatures preceding an upcoming slow slip event. **b** Slow slip timing from PNSN tremor logs (gray shaded areas), and smoothed (over 10 days) cataloged tremor rates in Southern Vancouver Island from the PNSN tremor catalog for comparison.

but with outlier values removed, which makes them more robust to signals not of interest for our analysis (such as earthquakes) and to potential sources of noise. A more extensive description of the features used can be found in the "Methods" section. These daily features are then averaged within a time window. Anomalous data points are detected within each window and removed before averaging. The results shown in Fig. 3 use a time window of 3 months (i.e. features are averaged over 90 days), but our methodology is robust to changes in the window size (see Supplementary). Each window is indexed by its latest day: the value of the features over the 3 months considered is associated with the last day of the window, to ensure that the analysis is made using only past data. Two successive time windows are offset by one day, and therefore contain 89 days in common. The averaged features over these time windows are used as input to

the ML algorithm. In the following, 'seismic features' will refer to those features averaged over a time window.

We apply a supervised ML approach to assess whether continuous seismic waves carry the potential signature of an upcoming slow slip failure. We assess whether a given time window of the continuous seismic data can be used to find signatures of impending failure for the next slow slip event. In the training phase, the algorithm takes as input the seismic features calculated from the first (contiguous) 50% of the seismic data (training set), and attempts to find the best model that maps these features to the time remaining before the next slow slip event (label or target). Details on how we build the model can be found in the "Methods" section.

Once a model is trained, it is evaluated on data it has never seen—the remaining (contiguous) 50% of the data (testing set). It

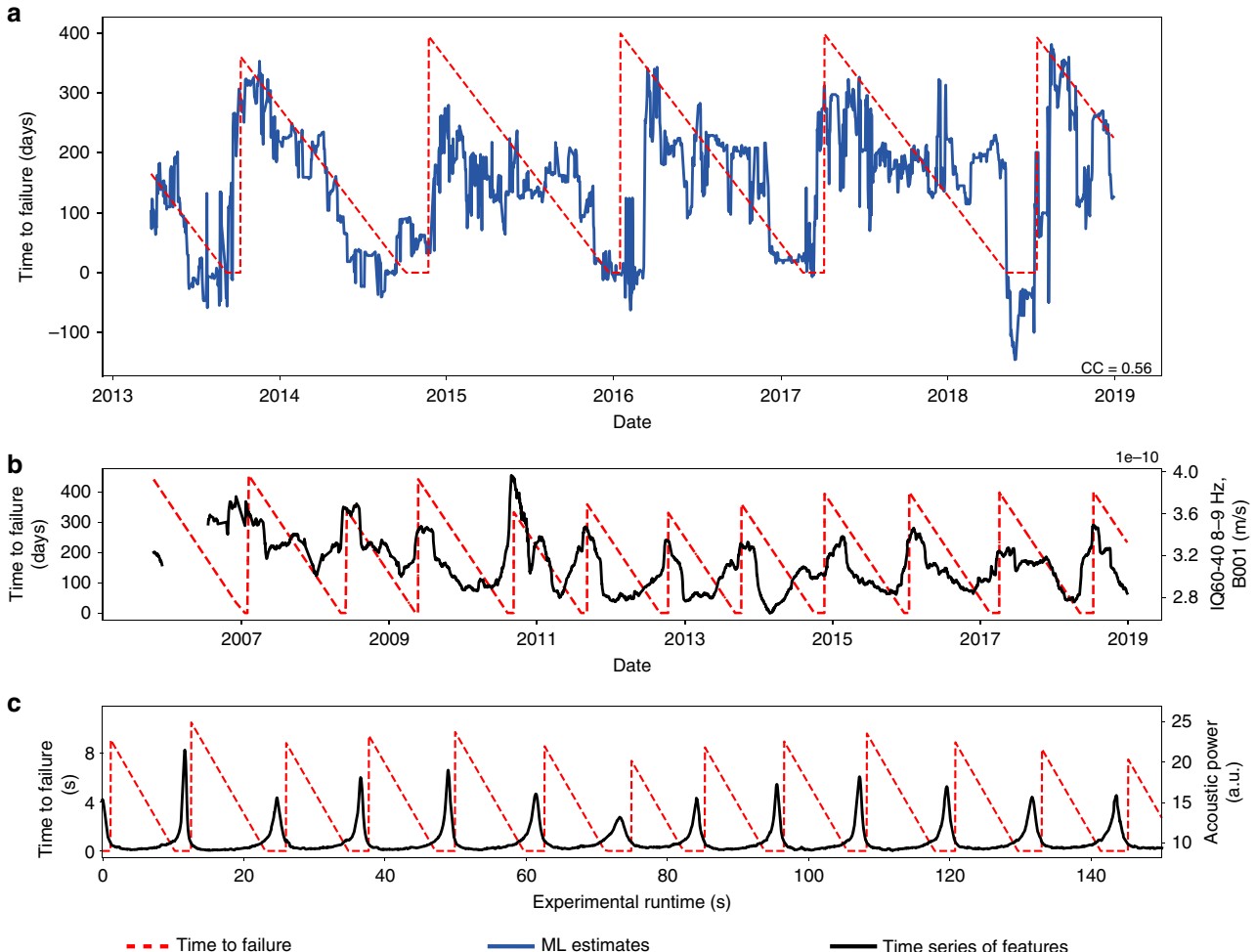

**Fig. 3 Slip timing estimations, patterns identified by our model, and comparison to shear experiments in the laboratory. a** ML estimates in testing of time to failure (in blue), and ground truth time to failure (in red) from the PNSN tremor logs. For most events, estimates are close to zero for several weeks preceding the rupture. **b** The most important feature identified by our model plotted against time, for the best stations (B001), for time window intervals of 3 months (black curve). This energy-based feature shows clear patterns with respect to the time remaining before the next slow slip event (left axis). **c** Shows the best statistical feature found in laboratory slow slip experiments for comparison (acoustic power, right-hand vertical axis). The best features in the laboratory **c** and Cascadia **b** are related to the energy of the seismic waves, and appear to follow similar patterns, with a progressive increase in amplitude that peaks towards the end of each slow slip event.

is important to note that in the testing phase, the model only has access to the seismic features calculated from the continuous seismic data, and has no information related to slow slip timing (the label). In the testing phase, the label is used exclusively to measure the quality of the model's estimates, i.e. how close these estimates are compared to the true label values obtained from PNSN Tremor Logs. If the model is able to estimate an imminent failure from seismic data it has never seen before, then it means that one or several features carry the signature of an impending slow slip event. We use Pearson's correlation coefficient (CC) as the evaluation metric, to compare the output of our model to the true test label values.

We rely on gradient-boosted trees algorithms that are relatively transparent in their analysis in contrast to many other methods[33]. These algorithms can be probed to identify which features are important in the model predictions, and why. Identifying the important statistical features allows us to make comparisons with laboratory experiments, and gain insight into the underlying physics.

**Estimating slip failure times from continuous seismic data.** Estimations of the time remaining before the next slow slip event

on the testing set are shown in Fig. 3. Figure 3a shows the ML time to failure estimations using data from station B001 (in blue), and the measured time remaining before the next slow slip event (ground truth, dashed red line). This ground truth can be understood as a countdown to the next slip, and is equal to zero whenever the PNSN Tremor Logs reported an ongoing slow slip. Each point of the model estimations of slip timing (blue curve) is made from seismic features from the three preceding months.

The seismic data long before failure and during failure appears very different to the trained model, that easily distinguishes between these two extreme cases in all the examples in our testing set. Estimations far from failure (for large values of time to failure) are noisy and not very accurate, but are of lesser interest for our analysis. They are often characterized by a long plateau during which the estimations remain relatively constant, suggesting that none of our seismic features follow patterns in the early stages of the cycle that are informative of the upcoming rupture. Interestingly, a similar but shorter plateau can be observed in laboratory slow slip events as well[13], especially following events of larger magnitudes. These observations suggest that slow slip events may be followed by a fundamentally unpredictable phase—maybe due to the local re-arrangement of

the system following the preceding failure. Other possible explanations could be that early signals are too small to be perceived amongst seismic noise, or that our selected set of features does not allow the algorithm to capture the early evolution of the system.

When close to failure (in the weeks preceding the rupture, when the time to failure approaches zero), the ML estimations are close to zero for most slow slip events—with the exception of the 2018 slow slip. Each point on the blue estimation curve in Fig. 3a is derived from a single time window of seismic data (built from the previous 90 days), that does not include any information from the rupture itself. Thus, the results suggest that at times closer to the end of the slow slip cycle, a snapshot of continuous seismic waves is imprinted with fundamental information regarding the upcoming failure of the system. The fact that estimations of failure times tend to drop abruptly also suggests that the evolution towards failure is less smooth than in the laboratory, and may be characterized by a sudden transition instead.

For most events, the signatures of impending failure can be observed weeks to months before the rupture. Furthermore, they are sufficiently similar from one cycle to the next for a model trained on past data to recognize them several years (cycles) afterwards. Whether the system remains stable enough for this exercise to hold over long periods of time is an open question—if more data was available, it might be the case that re-training on more recent data would be required after a while.

Note that because the events are most often separated by 13 or 14 months, the model cannot rely on seasonal signals to make its estimations. To prove this is the case, we show that a seasonal sinusoidal cannot lead to good estimations (i.e. simply knowing the time of the year, the ML model cannot make good predictions —see Supplementary). Therefore, the seismic data analyzed contains identifiable seismic patterns that match a slow slip cycle of 13–14 months and are independent of the season.

Because we choose to rely on transparent ML algorithms, we can identify the most important features used by our model to make its estimations of failure times, and therefore make comparison with laboratory experiments. We find that the best features identified in Cascadia follow very similar patterns compared to those identified in the laboratory. In the case of laboratory slow slip events, the most important feature by far for forecasting failure time is the seismic power[13,34], shown in Fig. 3c. In the case of the Cascadia subduction zone we rely on interquantile ranges, closely related to the root of the seismic power but with outlier values removed (these outliers are more likely to come from anthropogenic noise and/or local and distant earthquakes). Figure 3b illustrates that seismic energy follows

similar patterns in the laboratory and Cascadia: (i) a progressive increase, especially when failure approaches, with peaks in energy reached toward the end of the slow slip (as the slipping phase of the cycle emits larger seismic amplitudes than the loading phase, due to intense episodes of tremor within the frequency bands considered); and (ii) an often abrupt decrease within each slip cycle, towards the end of an event. Cycles in Cascadia are clearly apparent, but much noisier. This is likely due to background noise and to the fact that slow slip events occur over a very large region and might not be at the same stage everywhere.

In the laboratory, these signals carry fundamental information regarding the frictional state of the system. The fact that a similar behavior can be observed in Cascadia, with energy patterns matching the 14-month slow slip cycle, may also provide indirect information regarding the evolution of friction and slip rate at much larger scale. In simulated data, the origin of the seismic signal is due to the evolution of the force network in the granular gouge[35,36]. In the Cascadia subduction zone, we can posit that part of this energy is emitted from a large number of asperities located on the fault interface. This strong resemblance suggests that some of the frictional physics may scale from the laboratory to subduction in Cascadia, bringing additional evidence for the self-similarity of slow slip nucleation and rupture, in the laboratory and in the field.

**An exponential build-up in seismic energy preceding failure suggests a long nucleation phase for slow slip events in Cascadia.** Stacking the evolution of our main feature for all cycles shows that seismic power within these tremor frequency bands increases exponentially as the fault approaches failure. Figure 4 shows the density distribution of the datapoints for all cycles considered (including events from both the training and the testing set). In the laboratory, we observe a clear exponential growth of the acoustic power, starting about 5–7 s before failure, for an interevent time of ca. 10 s. We interpret this increase in acoustic power as the signature of a growing slip instability. In the case of the Cascadia slow slip events, stacked data highlight a comparable increase in seismic power emerging ~100 days before failure.

This exponential build up observed preceding failure is consistent with laboratory experiments focusing on earthquake nucleation[37]. During the nucleation of a slip instability, slip rate and rupture size both grow exponentially until the rupture has reached a size allowing for dynamic rupture propagation[38]. The observed exponential build-up in seismic energy in tremor frequency bands can be explained by an exponential growth of

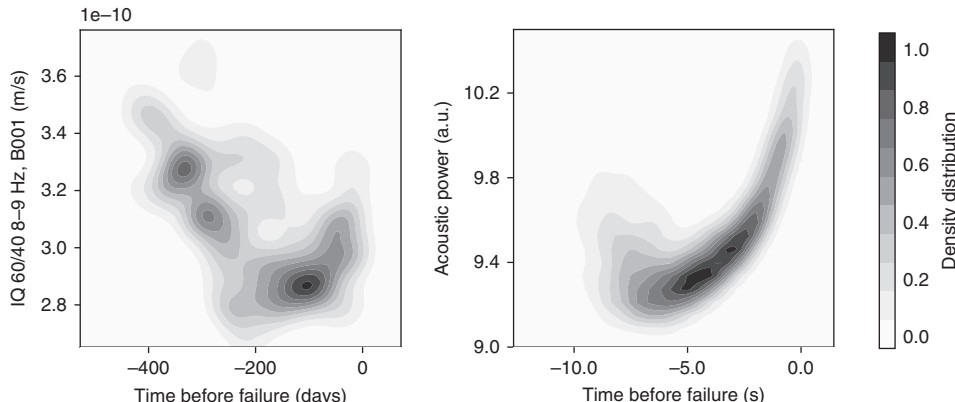

**Fig. 4 Density distribution.** Distribution of features with respect to time before failure, in between slip events—energy increases exponentially when failure becomes close. Right: laboratory slow slips. Left: Cascadia slow earthquakes.

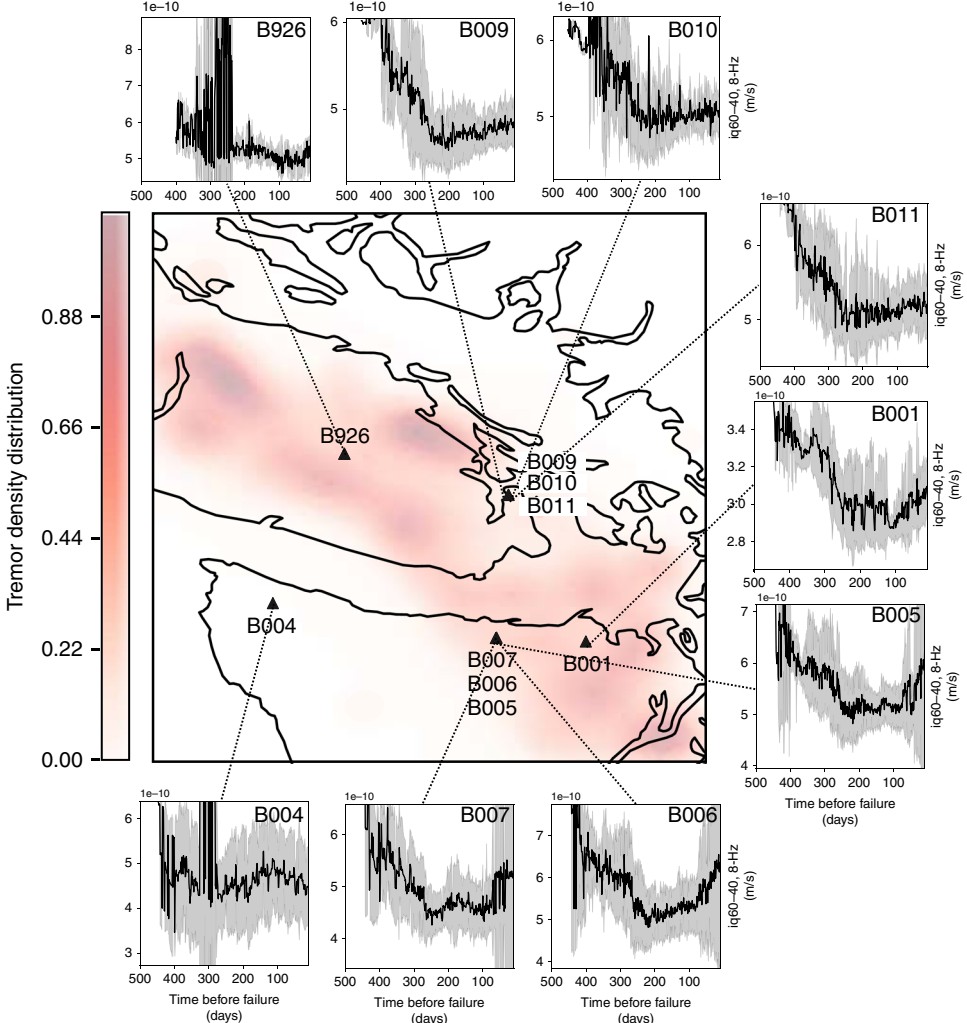

**Fig. 5 Behavior on all stations.** Mean (black curve) and standard deviation (gray shade) of our feature (seismic power in a tremor frequency band) for all stations considered, in-between slow slip events. For most stations, an increase in seismic power can be observed as failure approaches. This increase is particularly marked for the stations further inside the slowly slipping region. The only station outside of the slow slip area (B004) is also the only station to not exhibit such an increase in seismic power.

either the size or the slip rate (or both) of the nucleation phase of a slow slip event. In particular, we know from recent work that the same seismic features map accurately to the slip rate in both the laboratory[13,39] and in Cascadia[20]. Therefore, our results suggest that slow slip often begins with an exponential acceleration on the fault, that can be small enough to not be captured in cataloged tremor.

The evolution of stacked seismic power early in the slip cycle is characterized by a decreasing trend in both Cascadia and the laboratory. Part of this trend might come from the re-arrangement of the system after failure, although the underlying drivers remain unclear. In Cascadia, the observed signals appear to be mostly driven by tectonics (with clear 14-month patterns), but superposed with low-amplitude seasonal fluctuations.

The increase in seismic power preceding failure can also be observed on other seismic stations (Fig. 5), in particular for the stations further inside the slowly slipping region (B001, B005, B006, B007, and B926 to a lesser extent as is appears noisier). Among these stations, B001 is characterized by the smallest standard deviation close to failure, which explains why it is the most useful station for our ML analysis (see Supplementary). Stations closer to the border of the area that experiences slow slip still show an increase, but not as pronounced (B009, B010, B011).

Interestingly, the only station analyzed that is located outside of the slow slip area (B004) is also the only station that does not show any increase preceding the rupture.

## Discussion

We interpret the observed build-up in seismic noise within tremor frequency bands preceding failure as the signature of an increase in the number and intensity of low-amplitude tremors, too small to appear on several stations and be cataloged with array-based techniques. In our recent work[40], we showed that a neural network can detect many more tremors compared to the catalog it has been trained on. The number of tremors detected on surface stations in the same area by neural network seems to accelerate around 100 days before the rupture as well (see Supplementary), although the behavior preceding failure is not as clear as in the continuous seismic noise (from borehole stations). This is a strong argument in favor of our signal being driven by underlying tremor.

Because the signatures of upcoming failure can be recognized by our trained model many years afterwards, our results argue in favor of the predictability of slow slip rupture. If indeed slow slip events share a strong resemblance with earthquakes, long-term signa-tures of growing slip instabilities might be measurable preceding

destructive earthquakes as well. This hypothesis is in line with studies of the nucleation of large interplate earthquakes[41], and with recent work on the detection of small-amplitude foreshock activity in California[42,43] showing that foreshock activity may be observable preceding a significant fraction (i.e. 30% to 50% according to the analysis) of earthquakes with magnitude >4. Finer detection of small tremors and foreshocks may ultimately illuminate a long nucleation phase for large earthquakes, and pave the way for better constraints on rupture timing.

## Methods

**Seismic data and features used to build the models**. We rely on seismic data from the the Plate Boundary Observatory[32], from mid-2005 to late 2018. We build on our previous work related to the estimation of the GPS displacement from seismic data in Cascadia to create our features. From this previous work, we know that the best features that map to GPS displacement are intermediate inter-quantile ranges within the 8–13 Hz band[44].

The data processing is as follows:

(i) We correct for the instrument gain, for all waveforms.

(ii) We clip the signal at 5E−7 m/s, to limit the impact of earthquakes and anthropogenic noise on the analysis, and focus on low-amplitude signals.

(iii) For every station and every day we then compute the 40–60 and 25–75 inter-quantile ranges, for the following frequency bands: 8–9, 9–10, 10–11, 11–12, 12–13 Hz. Because these features discard the values of the waveforms below and above a certain percentile, they are robust to outlier values.

(iv) Once these features are built, anomalous feature datapoints are detected for each day and removed using Isolation Forests[45], with the automatic contamination threshold.

The features are then averaged over a time window and used as input to the ML models (the paper shows results with 3-month windows). The difference between the features' values at the beginning and the end of a given time window is also given as input to the algorithm. This leads to a total of $2*5*2 = 20$ features for each station. We build different models for each of the stations; combining stations or stacking station data does not lead to improved performance. Models built for each seismic station can be found in the Supplementary.

**Model construction and model hyperparameters**. We rely on the XGBoost library[46] for the gradient boosted trees' regression, shown in Fig. 2 of the paper (and for results presented below). The problem is posed in a regression setting. Model hyperparameters are set by five-fold cross-validation, using Bayesian optimization (skopt library[47]). The associated hyper-parameters for the regression in Fig. 3 of the main text (B001 station, 3-month windows) are the following: colsample_bytree = 0.90000000000000002, learning_rate = 0.20419758706505553, max_depth = 8, min_child_weight = 5, n_estimators = 947, reg_alpha = 100, reg_lambda = 100, subsample = 0.98999999999999999.

**Model evaluation**. We use Pearson's CC as evaluation metric. The Pearson coefficient is a measure of the strength of the linear relationship between the predicted and ground truth values. For a perfect positive linear relationship, the Pearson coefficient is 1.

## Data availability

The seismic data used was obtained from the Plate Boundary Observatory[32], and is publicly available in the IRIS DMC repository (https://ds.iris.edu/ds/nodes/dmc/earthscope/pbo/). The data can be accessed through the Obspy open source library, or other seismic query open-source softwares.

## Code availability

The code is based on two open-source libraries, XGBoost[46] and Scikit-Optimize[47], that are available online. We are currently building an open-source implementation of the code, that will be published online in the months to come.

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

## Acknowledgements

We thank C. Duverger and A. Schubnel for useful discussions and comments. We thank C. Marone, C. Bolton, and J. Rivière for the laboratory data. C.H. and B.R.-L. were supported by Institutional Support (LDRD) at Los Alamos. C.H.'s work was also supported by a joint research laboratory effort in the framework of the CEA-ENS Yves Rocard LRC (France), and both C.H. and R.J. were funded by the European Research Council (ERC) under the European Union's Horizon 2020 research and innovation program (Geo-4D project, grant agreement 758210). P.A.J. and CH were supported by DOE Office of Science (Geoscience Program, grant 89233218CNA000001).

## Author contributions

C.H. and B.R.-L. conducted the analysis. R.J. helped analyzing the results and provided expertise on earthquake nucleation. P.A.J. supervised the project. All authors contributed to writing the manuscript.

## Competing interests

The authors declare no competing interests.
