## [Peer Review File · Nature Communications]

REVIEWERS' COMMENTS:

Reviewer #1 (Remarks to the Author):

Hulbert et al. trained a Machine Learning algorithm to predict the time before the next Slow Slip Event (SSE) in Cascadia. The training is performed on 3-month-long continuous seismic time series recorded on nearby seismic stations. The labels are defined based on independent inference of tremor occurrence from the Pacific Northwest Seismic Network (PNSN) catalog. Once trained the algorithm gives impressively accurate prediction of the time before the next SSE using data it has never seen before. This study is remarkable and has profound implications on our current understanding of fault behavior, which will be of great interest to a broad community of people interested in earthquake problems. It is well written and pretty much ready for publication as it is. I am listing a few minor comments / suggestions below that may help the authors to clarify some points and further broaden the impact of the article. I look forward to seeing this manuscript published.

The figures would benefit to be larger. In particular Figure 3A should really take the full-page width. It is the main result of the study. It would be nice to be able to see the details of the prediction curve, in particular how long before the SSEs the algorithm is predicting their occurrences.

Figures 2A, 2B, 3B, 3C and 5 would also benefit from being larger (wider).

Abstract: I noticed you don't mention Machine Learning at all in the abstract. It is an important point of the study and probably should be mentioned in the Abstract. Also, you might want to define seismic power (in the abstract or later).

L42: I'm not sure what you mean here by "mapping"

L73: "whole Cascadia region"

Figure 2A: could you detail this figure a bit more? Down-sampled to which sampling rate? Clipped to what value?

L105: "Machine Learning (ML)" (I don't think you defined ML before)

L128: Remove coma. Move citation to the end of the sentence.

Figure 3A: how do small SSEs (visible as moderate tremor bursts in the PNSN catalog) fit in that picture? Even though the algorithm was not trained to detect such kind of event I would be curious to see if they correspond to local drops in the prediction curve. It may not work as the algorithm was not design for this task but if it does, it would be interesting to show.

L180: you might want to define seismic power.

L184: you can remove "the fact" and "both"

L186: Figure 3B shows a value averaged on the 90 days before right? This probably explains why the peaks in energy are reached toward the end of the slow slip (there is a 45-day offset).

L200: do you mean earthquake nucleation?

Figure 4: Starting the time axis at -200 days would probably strengthen your point. Also, show the density color scale.

L228: it's not clear what "This" refer to at the beginning of a paragraph.

L233: a "tremor event" is not really a thing. Tremor refers to a seismic signal which is thought to emerge from the superposition of many low-frequency earthquakes (LFEs) (Shelly et al., Nature, 2007).

L239-244: Could you plot the daily number of tremor detections identified in your previous study on a larger, wider version of Figure 3A. The relationship would be interesting to see on a figure.

L251-253: this observation has been questioned (Van den Ende and Ampuero, GRL, 2019)

Supplements:

You forgot someone in the author list.

P1-L4: "widow" -> "window"

Figure S1: Looking at this figure, I can't help but wonder why you didn't use the 110-day windows. It seems to work significantly better with that duration. Any thought why? Why did you stick with 90 days?

P2-L9: "station" -> "stations"

P4-Lend-1: "We rely on GPS data for the station ALBH located on Vancouver Island" -> "We use the GPS station ALBH, which is located on Vancouver Island"

P5-L1: "the total horizontal displacement (E+N)" -> "the horizontal displacement projected on the northeast direction (E+N)"

Figure S3B: This figure should be a lot larger

P7-L1: "S6" -> "S5". "figure" -> "figure S5"

P9: see my comment on "tremor events"

I hope this helps.

Quentin Bletery

References:

Shelly, D. R., Beroza, G. C., & Ide, S. (2007). Non-volcanic tremor and low-frequency earthquake swarms. *Nature*, 446(7133), 305-307.

Van den Ende, M. P. A., & Ampuero, J.-P. (2020). On the statistical significance of foreshock sequences in Southern California. *Geophysical Research Letters*, 47, e2019GL086224.

Reviewer #2 (Remarks to the Author):

Review of Hulbert et al, "An Exponential Build-up in Seismic Energy Suggests a Months-Long Nucleation of Slow Slip in Cascadia", submitted to *Nature Communications*.

This paper builds on previous work from the same group, developing cutting edge machine learning studies of seismic signals from laboratory and natural faults, which is among the most

exciting work in fault mechanics and earthquake seismology at present. In their most relevant previous paper in Nature Geoscience "Continuous chatter of the Cascadia subduction zone revealed by machine learning", by Rouet-Leduc et al., Nature Geoscience 2019, they were able to predict GPS (surface deformation) signals by training supervised learning methods on seismic data. In their first ground-breaking study, they analyzed experimental data with supervised learning methods and were able to predict the time to the next stick-slip event in the sample based only on a few parameters (unlike a neural network, their decision tree-based methods indicate what kinds of data have the most predictive power). In this paper, they setup an analysis of continuous seismic data to predict onset time for slow slip events, based on pre-identified large, coherent tremor events defining slow slip (Episodic Tremor and Slip, ETS) events, compiled into a catalog. They identify a fascinating similarity in the time-evolution of seismic energy between the laboratory fault and Cascadia, and demonstrate the self- similarity of the evolution in noise amplitude at vastly different length scales, shown in Figure 4 of this manuscript. It is remarkable and shows significant evidence for the kinds of consistent rupture precursors in natural data, akin to those that they identified in the lab. The next question, of course, is if such precursors can be identified in "normal" earthquakes that can cause major damage. This paper represents a significant and clear contribution towards that aim. I encourage its publication, with just some minor comments and questions for clarification.

Comments to the Authors:

lines 22-23: "...increases pore pressure, hence inhibiting brittle failure." An increase of fluid in rock can inhibit fracture/brittle failure by damping crack tip stresses relative to unsaturated conditions at relevant temperatures, but increased pore pressure should enhance brittle failure at such conditions. (Fig 2 of <https://agupubs.onlinelibrary.wiley.com/doi/pdf/10.1002/2015JB012047>) Clarify?

lines 67-72: PNSN Tremor Logs and PNSN tremor catalog are different things? Looking at the "Tremor Logs" on the PNSN website it looks like a blog, not a catalog. The reference (22) seems to focus on one ETS event.

lines 80-82: "GPS displacement and nearby tremor are measured locally, whereas seismic data may capture signatures of slipping segments located farther away." Confusing terminology here: "nearby tremor" is also "seismic data", right? Or is this referring to located tremor events in the catalog (some sort of derived/extracted data) rather than the tremor in the seismic data itself? Please clarify.

line 94: Is there clear dispersion in the signals, shifting energy up or down in frequency with time during one tremor event, that might be apparent in these narrow band pass results? Probably not relevant to the analysis-- just curious.

line 99: It would be good to describe the nature of the features a bit more here (and this is done in the Methods, not the Supplement as stated here), so that readers unfamiliar with the groups previous work can get a better sense for the methods and how to think about the resulting patterns. In some ways, this choice in the ML analysis is at the heart of the method, with a lot of experience embedded, that readers should know a bit more about without having to go dig (even to the Methods, though the detail should be there, not the Supplement, which gets lost).

lines 102-106: This description is confusing. Three month windows with one day overlaps? so you take 90 days worth of single-day statistics and then average them and report that value for the day (which is the last day in the window-- so there is only memory, no future knowledge of course!), and then slide the 90 day window by one day? Are the performance metric values shown in Fig. S1 significant? 110 is much better than 100? And interestingly, it looks like 120-day windows is rougher than 100-day windows.

Fig 3 Caption: "Failure times" in blue? It is not "time to failure" predicted? and "zero several"="zero to several"?

line 181-182: Outlier values removed from the waveform? things that remain after the narrow band filtering and clipping-- Or outliers in the statistics of the signal? I think it would be better to explain this in methods summary paragraph with the line 99 comment.

Fig 4. Great figure ! This may be a figure that is studied for a while to come !

lines 221: Underlying drivers of the downward trend? Is it not just annealing/healing-- faults settling into their new configurations (as you say) and getting stiffer? But what determines the location of the minima, and the change to a positive slope? A point at which any continued fault healing doesn't make the faults any stiffer, as background elastic energy builds up?

lines 224-226: What might cause the seasonal variations in the 8-13 Hz range? Wind-driven oscillations ?

237-241:

Isn't the timing of the onset of a "failure" event dependent on the threshold level chosen to define an "episode" ? Seems like the DNN detection and noise analysis should converge at some point-- Tremor catalogs have a detection threshold, which is lowered by a DNN (Rouet-Leduc et al). In the fictional case (or the laboratory) that there could be a seismometer close to the fault surface, detection of events would be far below the "noise" level at the surface (i.e. the paths from sources to seismometer has a blurring/smearing effect). The definition of the onset of an event may change with detection improvements, based on the kind of analysis presented here, but what about the physics of the peak related to the character of the ramping up of the noise amplitude towards the peak?

lines 244: Related question: What does "our signal being driven by underlying tremor" mean ? Your signal *is* underlying tremor, no? Or maybe clarify what "This" refers to ?

To the editors:

All that said, this is a very interesting paper; these are only minor modifications for clarifications, and questions of interest.

I strongly support publishing it!

--Ben Holtzman

Reviewer #3 (Remarks to the Author):

Dear Editor,
Dear Authors,

Please find below my comments on the manuscript entitled "An Exponential Build-up in Seismic Energy Suggests a Months-Long Nucleation of Slow Slip in Cascadia" submitted to Nature Communications. The authors have produced a very clear manuscript. The scientific ideas are exposed in a pedagogic fashion with good-quality results. The authors also make several connections with laboratory experiments, which strengthen the arguments and will be of interest to a broad audience. Therefore, I consider this study deserves being published in Nature Communications. Please consider below the list of minor comments and suggestions to the authors that may help to strengthen the paper.

- Figure 1: there is a typo in the colorbar label (Density distribution). In the text, the authors probably omitted to refer to Figure 1B (line 71) and may want to refer to Figure 1A only in line 56.
- Discussion about the visible plateaus on the predicted time to failure (lines 148 to 151): while the two discussed points are definitely possible, an additional reason would be that the set of selected features (or data processing) does not allow to capture the early times. If the potential seismic signatures are buried in seismic noise, this comment is redundant to the already-mentioned argument. But if the signature lies outside the feature space or in a different time scale/frequency band, this may also prevent from correctly predicting the time to failure.
- General comment about negative time-to-failure: would you think it would make sense to force the predicted time-to-failure to remain positive? This raises the question of how much a priori one should bring to the procedure, but with this strong additional constrain, could it be possible that the algorithm makes better predictions?
- Line 114: why did you split the training and testing dataset in a continuous way instead of shuffling it? Is it to ensure that both training and testing sets contain approximatively an equal amount of full cycles? Could it be possible to see also the prediction on the training set to have an idea of the generalization error?
- Figure 4: adding colorbars could help have an idea of the contour contrast.

Many thanks to the reviewers for their thoughtful reviews, and for their comments that were very helpful to improve and clarify the manuscript. We tried to address all comments as well as possible, and describe the changes below. We worked in particular on improving the figures, by enlarging most of them to make them easier to visualize, adding colorbars, or dividing subplots. The Supplementary was modified extensively to meet formatting requirements, but the content remains mostly unchanged, except for modifications detailed below.

In what follows, the reviewers' comments are in grey, and our responses are in black. Changes in the manuscript and supplementary are indicated in bold.

Reviewer #1 (Remarks to the Author):

Hulbert et al. trained a Machine Learning algorithm to predict the time before the next Slow Slip Event (SSE) in Cascadia. The training is performed on 3-month-long continuous seismic time series recorded on nearby seismic stations. The labels are defined based on independent inference of tremor occurrence from the Pacific Northwest Seismic Network (PNSN) catalog. Once trained the algorithm gives impressively accurate prediction of the time before the next SSE using data it has never seen before. This study is remarkable and has profound implications on our current understanding of fault behavior, which will be of great interest to a broad community of people interested in earthquake problems. It is well written and pretty much ready for publication as it is. I am listing a few minor comments / suggestions below that may help the authors to clarify some points and further broaden the impact of the article. I look forward to seeing this manuscript published.

Thanks a lot for these supportive words and constructive comments! We have done our best to include them, and improved the figures as suggested.

The figures would benefit to be larger. In particular Figure 3A should really take the full-page width. It is the main result of the study. It would be nice to be able to see the details of the prediction curve, in particular how long before the SSEs the algorithm is predicting their occurrences.

Figures 2A, 2B, 3B, 3C and 5 would also benefit from being larger (wider).

We re-worked all of the afore-mentioned figures, in particular Figure 3, to make them wider.

Abstract: I noticed you don't mention Machine Learning at all in the abstract. It is an important point of the study and probably should be mentioned in the Abstract. Also, you might want to define seismic power (in the abstract or later).

We added a sentence to the abstract to mention that a large part of our analysis relies on machine learning :

« We rely on machine learning algorithms to infer slow slip timing from statistics of seismic waveforms. »

We also added a definition of seismic power in line 43:

« we define seismic power as the average of seismic energy per unit of time, i.e. the squared measured ground velocity per unit of time. »

L42: I'm not sure what you mean here by "mapping"

We changed « mapping » to « proportionality » in order to clarify the sentence:

«The proportionality between seismic power and surface displacement enables a quantitative characterization of slow slip events from seismic data. »

L73: "whole Cascadia region"

Indeed this formulation is less heavy – we modified the text accordingly.

Figure 2A: could you detail this figure a bit more? Down-sampled to which sampling rate? Clipped to what value?

The data was only down-sampled for the plot, as it wasn't possible to plot many years at 100Hz (the data used in the analysis is not down-sampled). The data is clipped before computing the features, at $5E-7$ m/s. We modified the caption of Figure 2 to clarify these points:

« Down-sampled (solely for visualization purposes) and clipped (at $5E-7$ m/s) continuous seismic waves for one station analyzed (B001). »

L105: “Machine Learning (ML)” (I don't think you defined ML before)

The term is introduced in line 68:

« Supervised machine learning (ML) used in the work described below, requires robust training and testing sets including many slip events. »

L128: Remove coma. Move citation to the end of the sentence.

Thanks! We changed the sentence accordingly.

Figure 3A: how do small SSEs (visible as moderate tremor bursts in the PNSN catalog) fit in that picture? Even though the algorithm was not trained to detect such kind of event I would be curious to see if they correspond to local drops in the prediction curve. It may not work as the algorithm was not design for this task but if it does, it would be interesting to show.

This is a very good question. It is difficult to answer however, due to the temporal smoothing used for the analysis. Typically smaller SSEs do not last for a very long time, whereas our continuous seismic waves are smoothed over large windows, which makes this potential connection difficult to visualize. We added a figure in the Supplementary (Figure S10) to compare the tremors detected by neural network to the model's estimates.

L180: you might want to define seismic power.

As mentioned above, we added a definition in line 43.

L184: you can remove “the fact” and “both”

We changed the sentence accordingly.

L186: Figure 3B shows a value averaged on the 90 days before right? This probably explains why the peaks in energy are reached toward the end of the slow slip (there is a 45-day offset).

Indeed, one point in Figure 3B corresponds to the average feature over the last three months.

However, because each window is only offset by one day, the behavior (increase or decrease of the feature) between two successive windows is only dictated by the last day considered in the averaging. Therefore we do not believe that the peak in the feature is offset due to the 3-month smoothing. Interestingly, in the lab (with much shorter windows compared to the length of the slow slip cycle, and therefore a smaller impact of the smoothing), the peak also occurs towards the end of the event.

L200: do you mean earthquake nucleation?

Here we refer to slow slip nucleation. The idea is that both slow slip rupture and nucleation seem to share very similar patterns regardless of scale (lab versus Cascadia), which argue in favor of self-similarity. We modified the text as follows:

« This strong resemblance suggests that some of the frictional physics may scale from the laboratory to subduction in Cascadia, bringing additional evidence for the self-similarity of slow slip nucleation and rupture, in the laboratory and in the field.»

Figure 4: Starting the time axis at -200 days would probably strengthen your point. Also, show the density color scale.

We believe that it may be better to keep all the data in the figure, as we discuss the downward trend observed early in the slip cycle (days -400 to -200) in the manuscript. As an alternative, we enlarged the figure a bit so that the behaviour before failure is more clearly visible. We also added a color scale to the figure (as well as to the similar figures S5 and S9 of the Supplementary).

L228: it's not clear what "This" refer to at the beginning of a paragraph.

We modified the text to clarify this sentence:

« The increase in seismic power preceding failure can also be observed on other seismic stations (Figure 5), in particular for the stations located right above the slowly slipping region (B001, B005, B006, B007, and B926 to a lesser extent as is it noisier). »

L233: a "tremor event" is not really a thing. Tremor refers to a seismic signal which is thought to emerge from the superposition of many low-frequency earthquakes (LFEs) (Shelly et al., Nature, 2007).

We replaced « tremor events » by « tremors » for all the occurrences in the manuscript.

L239-244: Could you plot the daily number of tremor detections identified in your previous study on a larger, wider version of Figure 3A. The relationship would be interesting to see on a figure.

We added a new figure in the Supplementary (Figure S10) to show the evolution of detected tremors vs the machine learning estimates from Figure 3A. As mentioned above, due to the large smoothing (3 months) of the contiguous seismic data, it is not obvious to match drops in the model's predictions to short-term bursts of tremor. However there may be a connection in some cases, visible even with the smoothing: for example in late 2014, it may be the case that our model estimates failure earlier than for other years because of many small tremor bursts occurring.

L251-253: this observation has been questioned (Van den Ende and Ampuero, GRL, 2019)

We modified the conclusion to have it more nuanced, and included the reference:

« This hypothesis is in line with studies of the nucleation of large interplate earthquakes, and with recent work on the detection of small-amplitude foreshock activity in California showing that foreshock activity may be observable preceding a significant fraction (i.e. 30% given published works) of earthquakes magnitude greater than 4. »

We also added a new references (Bouchon et al., 2013) to this new version of the conclusion, as we now discuss large interplate earthquakes.

Supplements:

You forgot someone in the author list.

Thanks a lot for noticing!

P1-L4: "widow" -> "window"

Thanks! We fixed the typo.

Figure S1: Looking at this figure, I can't help but wonder why you didn't use the 110-day windows. It seems to work significantly better with that duration. Any thought why? Why did you stick with 90 days?

We wanted to put emphasis on the smallest window size that yields good results. In our perspective, the smaller the window is compared to the slip cycle, the better – as it proves that the extracted signal is more local in time (and a 90-day smoothing is already large compared to the 14-month slow slip cycle).

P2-L9: “station” -> “stations”

Thanks! We fixed it.

P4-Lend-1: “We rely on GPS data for the station ALBH located on Vancouver Island” -> “We use the GPS station ALBH, which is located on Vancouver Island”

We modified the text.

P5-L1: “the total horizontal displacement (E+N)” -> “the horizontal displacement projected on the northeast direction (E+N)”

We modified the text.

Figure S3B: This figure should be a lot larger

We modified Figure S3 (now Figure S4) to make the subplots bigger. We also changed the text describing the figure and the caption, to make them clearer and to reference explicitly the different subplots.

P7-L1: “S6” -> “S5”. “figure” -> “figure S5”

Thanks for noticing!

P9: see my comment on “tremor events”

We replaced all occurrences of “tremor events” by “tremors” also in the Supplementary.

I hope this helps.

Quentin Bletery

References:

Shelly, D. R., Beroza, G. C., & Ide, S. (2007). Non-volcanic tremor and low-frequency earthquake swarms. *Nature*, 446(7133), 305-307.

Van den Ende, M. P. A., & Ampuero, J. □P. (2020). On the statistical significance of foreshock sequences in Southern California. *Geophysical Research Letters*, 47, e2019GL086224.

Reviewer #2 (Remarks to the Author):

Review of Hulbert et al, "An Exponential Build-up in Seismic Energy Suggests a Months-Long Nucleation of Slow Slip in Cascadia", submitted to Nature Communications.

This paper builds on previous work from the same group, developing cutting edge machine learning studies of seismic signals from laboratory and natural faults, which is among the most exciting work in fault mechanics and earthquake seismology at present. In their most relevant previous paper in Nature Geoscience "Continuous chatter of the Cascadia subduction zone revealed by machine learning", by Rouet-Leduc et al., Nature Geoscience 2019, they were able to predict GPS (surface deformation) signals by training supervised learning methods on seismic data. In their first ground-breaking study, they analyzed experimental data with supervised learning methods and were able to predict the time to the next stick-slip event in the sample based only on a few parameters (unlike a neural network, their decision tree-based methods indicate what kinds of data have the most predictive power). In this paper, they setup an analysis of continuous seismic data to predict onset time for slow slip events, based on pre-identified large, coherent tremor events defining slow slip (Episodic Tremor and Slip, ETS) events, compiled into a catalog. They identify a fascinating similarity in the time-evolution of seismic energy between the laboratory fault and Cascadia, and demonstrate the self- similarity of the evolution in noise amplitude at vastly different length scales, shown in Figure 4 of this manuscript. It is remarkable and shows significant evidence for the kinds of consistent rupture precursors in natural data, akin to those that they identified in the lab. The next question, of course, is if such precursors can be identified in "normal" earthquakes that can cause major damage. This paper represents a significant and clear contribution towards that aim. I encourage its publication, with just some minor comments and questions for clarification. Many thanks for these encouraging words, and comments that helped us a lot to clarify the manuscript!

Comments to the Authors:

lines 22-23: "...increases pore pressure, hence inhibiting brittle failure." An increase of fluid in rock can inhibit fracture/brittle failure by damping crack tip stresses relative to unsaturated conditions at relevant temperatures, but increased pore pressure should enhance brittle failure at such conditions. (Fig 2 of <https://agupubs.onlinelibrary.wiley.com/doi/pdf/10.1002/2015JB012047>) Clarify?

Thanks for catching this! Here we meant 'dynamic failure'. Indeed in rock experiments, it appears that near the brittle/ductile transition regime (of interest in our analysis), increases in pore pressure tend to favor slow slip (cf reference from the link above). We modified the manuscript to clarify that our point refers to dynamic failure in the brittle/ductile transition regime, and added the aforementioned reference:

« At such depths, slow slip and tremor are thought to take place where temperatures drive dehydration of subducting material that increases pore pressure, inhibiting dynamic failure in the brittle/ductile transition regime ».

lines 67-72: PNSN Tremor Logs and PNSN tremor catalog are different things? Looking at the "Tremor Logs" on the PNSN website it looks like a blog, not a catalog. The reference (22) seems to focus on one ETS event.

The tremor logs are indeed web pages from the PNSN that analyze each event, based on tremor detections. There is one page for each slow slip event (the reference links to one of these pages). They provide a beginning and an end day for each event, based upon the tremor detections made by the institution, starting in 2003. We chose to rely on these timings because data from the PNSN tremor catalog itself only starts in late 2009, which would lead us to discard a third of the available

data. Figure 2(B) illustrates the slip timings from the tremor logs, versus smoothed tremor detections in Vancouver Island from the PNSN catalog, over the period considered (2005-2018). We added a paragraph in the Supplementary to describe them:

« The tremor logs, available on the PNSN website, are web pages that analyze each slow slip event, and provide a beginning and an end day for each event based upon the tremor detections made by the institution (starting in 2003). We chose to rely on these timings because they encompass the whole Cascadia area, and because data from the PNSN tremor catalog itself only starts in late 2009 - which would lead us to discard a third of the available data ».

lines 80-82: "GPS displacement and nearby tremor are measured locally, whereas seismic data may capture signatures of slipping segments located farther away." Confusing terminology here: "nearby tremor" is also "seismic data", right? Or is this referring to located tremor events in the catalog (some sort of derived/extracted data) rather than the tremor in the seismic data itself? Please clarify.

Indeed the formulation is not very clear! The main difference here between 'nearby tremor' and 'seismic data' is in terms of proximity of the tremor location: a seismic sensor will capture a tremor that occurs close by, but could also capture a relatively strong-amplitude tremor that is located outside of our area of interest. We modified « seismic data » by « seismic sensor » in the sentence, which we believe clarifies the meaning:

« GPS displacement is a local measurement, and so is the occurrence of local tremor, whereas seismic sensors may capture signatures of slipping segments located farther away. »

line 94: Is there clear dispersion in the signals, shifting energy up or down in frequency with time during one tremor event, that might be apparent in these narrow band pass results? Probably not relevant to the analysis-- just curious.

It is not obvious that this is happening. We actually tried to test this hypothesis a little while ago, by building another version of our algorithm where the features were gradients between frequencies, instead of frequency values. This version had a much worse performance than the model presented here. This doesn't mean that there is no dispersion – but at least, it appears to be much less informative.

line 99: It would be good to describe the nature of the features a bit more here (and this is done in the Methods, not the Supplement as stated here), so that readers unfamiliar with the groups previous work can get a better sense for the methods and how to think about the resulting patterns. In some ways, this choice in the ML analysis is at the heart of the method, with a lot of experience embedded, that readers should know a bit more about without having to go dig (even to the Methods, though the detail should be there, not the Supplement, which gets lost).

Yes, the features are described in the Methods section and not in the Supplementary - thanks for noticing! We added a description of the features in the main text:

« Building on our previous work, the features correspond to inter-quantile ranges of seismic data within tremor frequency bands (8 to 13Hz, by 1Hz increments). These features are representative of seismic energy within the usual tremor frequency bands, but with outlier values removed, which makes them more robust to signals not of interest for our analysis (such as earthquakes) and to potential sources of noise. A more extensive description of the features used can be found in the Methods section».

lines 102-106: This description is confusing. Three month windows with one day overlaps? so you take 90 days worth of single-day statistics and then average them and report that value for the day (which is the last day in the window-- so there is only memory, no future knowledge of course!), and then slide the 90 day window by one day? Are the performance metric values shown in Fig. S1 significant? 110 is much better than 100? And interestingly, it looks like 120-day windows is rougher than 100-day windows.

Yes, this is correct. One window corresponds to the average value of the features over 90 days, and is reported for the last day so that the algorithm never sees any data from the future. Two successive windows are offset by one day, so they have 89 days in common. This is done so that the algorithm has enough datapoints to learn from; significantly decreasing the overlap would drastically reduce the size of the database and hurt the model's performance.

Because the windows overlap so much, the contiguous train-test split is of particular importance in our study. If the split was random, the algorithm could just use these 89 common days and produce a very good estimate, albeit completely artificial. The contiguous split forces the algorithm to consider completely new datapoints, that do not have any days in common with examples seen in training.

We modified the text to clarify the procedure:

« The results shown in Figure 3 use a time window of 3 months (i.e. features are averaged over 90 days), but our methodology is robust to changes in the window size (see Supplementary). Each window is indexed by its latest day: the value of the features over the three months considered is associated with the last day of the window, to ensure that the analysis is made using only past data. Two successive time windows are offset by one day, and therefore contain 89 days in common. »

The changes in performance with respect to window size are mostly driven by how noisy the ML estimates are: even if the overall behaviour is the same, relatively small noisy fluctuations will impact the correlation metric. This is also why we chose to report the results with 90-day windows, and not those with the higher performance as measured by pearson correlation. Smaller window sizes are more interesting with respect to the physical analysis, regardless of the CC score.

Fig 3 Caption: "Failure times" in blue? It is not "time to failure" predicted? and "zero several"="zero to several"?

Yes indeed, it is time to failure! "zero several" has been changed to « zero for several », to clarify the meaning.

line 181-182: Outlier values removed from the waveform? things that remain after the narrow band filtering and clipping-- Or outliers in the statistics of the signal? I think it would be better to explain this in methods summary paragraph with the line 99 comment.

We mean that the inter-quantile ranges themselves, because they discard the values of the waveforms below and above a given percentage, are robust to outliers. We modified the Methods section to describe the processing more clearly, and with more details:

« The data processing is as follows:

- i) We correct for the instrument gain, for all waveforms.
- ii) We clip the signal at $5E-7$ m/s, to limit the impact of earthquakes and anthropogenic noise on the analysis, and focus on low-amplitude signals.
- iii) For every station and every day we then compute the 40-60 and 25-75 interquantile ranges, for the following frequency bands: 8-9, 9-10, 10-11, 11-12, 12-13Hz. Because these features discard the values of the waveforms below and above a certain percentile, they are robust to outlier values present in the waveforms.
- iv) Once these features are built, anomalous feature datapoints are detected for each day and removed using Isolation Forests, with the automatic contamination threshold. »

Fig 4. Great figure ! This may be a figure that is studied for a while to come !
Thanks a lot for these kind words!

lines 221: Underlying drivers of the downward trend? Is it not just annealing/healing-- faults settling into their new configurations (as you say) and getting stiffer? But what determines the location of the minima, and the change to a positive slope? A point at which any continued fault healing doesnt make the faults any stiffer, as background elastic energy builds up?

This is a very good question, to which we do not have an answer at this point – neither in the lab or in Earth. Hopefully future work will help us to understand this transition better!

lines 224-226: What might cause the seasonal variations in the 8-13 Hz range? Wind-driven oscillations ?

This is something that we have started to analyze. Wind, rainfall and hydrology, as well as temperature, are likely to have an impact on these seasonal variations.

237-241:

Isn't the timing of the onset of a "failure" event dependent on the threshold level chosen to define an "episode" ? Seems like the DNN detection and noise analysis should converge at some point-- Tremor catalogs have a detection threshold, which is lowered by a DNN (Rouet-Leduc et al). In the fictional case (or the laboratory) that there could be a seismometer close to the fault surface, detection of events would be far below the "noise" level at the surface (i.e. the paths from sources to seismometer has a blurring/smearing effect). The definition of the onset of an event may change with detection improvements, based on the kind of analysis presented here, but what about the physics of the peak related to the character of the ramping up of the noise amplitude towards the peak?

Yes, this is very true. We believe that the current analysis does lower the detectability of the slow slip rupture, and indeed the results shown here are in good agreement with the tremor detected by the DNN (although in its current stage, the DNN analysis is still less informative than the continuous seismic analysis). This is encouraging, as it poses the question of whether similar detection improvements could enhance the detection of the onset of large earthquakes. In terms of physics, the analysis of both lab and field data point towards an acceleration in slip velocity, suggesting that the peak corresponds to the maximum slip acceleration reached during the rupture. All of our observations so far, in particular in the lab, are in favor of a pre-slip nucleation model where dynamic rupture is preceded by slow slip. Therefore, in the (far-fetched) hypothesis that such observations directly extend to earthquakes in Earth, the ramping up would correspond to faster and faster aseismic slip, and the peak would correspond to the dynamic rupture.

lines 244: Related question: What does "our signal being driven by underlying tremor" mean ? Your signal *is* underlying tremor, no? Or maybe clarify what "This" refers to ?

We believe that our signal is indeed underlying tremor, given that i) all the features are computed within tremor frequency bands, and ii) the acceleration ~100 days before failure appears to match the observed increase of tremor detections by DNN. However our analysis remains indirect, as we do not rely on detected tremors here, only on continuous signals. This explains why we added this caveat.

To the editors:

All that said, this is a very interesting paper; these are only minor modifications for clarifications, and questions of interest.

I strongly support publishing it!

--Ben Holtzman

Reviewer #3 (Remarks to the Author):

Dear Editor,
Dear Authors,

Please find below my comments on the manuscript entitled "An Exponential Build-up in Seismic Energy Suggests a Months-Long Nucleation of Slow Slip in Cascadia" submitted to Nature Communications. The authors have produced a very clear manuscript. The scientific ideas are exposed in a pedagogic fashion with good-quality results. The authors also make several connections with laboratory experiments, which strengthen the arguments and will be of interest to a broad audience. Therefore, I consider this study deserves being published in Nature Communications. Please consider below the list of minor comments and suggestions to the authors that may help to strengthen the paper.

Thanks a lot for these supportive and constructive comments. We tried to include them as well as possible.

- Figure 1: there is a typo in the colorbar label (Density distribution). In the text, the authors probably omitted to refer to Figure 1B (line 71) and may want to refer to Figure 1A only in line 56. Thanks for noticing the typo! We changed the label.

We also modified the text to refer explicitly to figures 1A and 1B:

« We analyze seismic data on Vancouver Island, Canada, where the Juan de Fuca oceanic plate subducts beneath the North American plate (Figure 1A). The quasi-periodic occurrence of slow slip events (approximately every 14 months) is manifested by the North American plate lurching southwesterly over the Juan de Fuca plate, generating bursts of tectonic tremor over the area (Figure 1B). »

- Discussion about the visible plateaus on the predicted time to failure (lines 148 to 151): while the two discussed points are definitely possible, an additional reason would be that the set of selected features (or data processing) does not allow to capture the early times. If the potential seismic signatures are buried in seismic noise, this comment is redundant to the already-mentioned argument. But if the signature lies outside the feature space or in a different time scale/frequency band, this may also prevent from correctly predicting the time to failure.

Indeed, this could very well be the case. We modified the text to include this possibility :

« Other possible explanations could be that early signals are too small to be perceived amongst seismic noise, or that our selected set of features does not allow the algorithm to capture the early evolution of the system. »

- General comment about negative time-to-failure: would you think it would make sense to force the predicted time-to-failure to remain positive? This raises the question of how much a priori one should bring to the procedure, but with this strong additional constrain, could it be possible that the algorithm makes better predictions?

The presence of time to failures below zero is a consequence of the choice of the algorithm used (gradient boosted trees are able to extrapolate to values unseen). Other families of algorithms that are unable to extrapolate could be used instead, but we believe that this is unlikely to improve the estimations. For visualization purposes and enhanced clarity, one could simply take the predictions below zero and replace them by zero.

- Line 114: why did you split the training and testing dataset in a continuous way instead of shuffling it? Is it to ensure that both training and testing sets contain approximatively an equal amount of full cycles? Could it be possible to see also the prediction on the training set to have an idea of the generalization error?

The contiguous train/test split is standard for time series, in order to avoid corruption of the testing set due to the auto-correlation in time of the phenomena analyzed. In our case, the use of moving windows makes this split particularly crucial: two contiguous windows are offset by one day, and therefore share 89 days in common. If the split was random, it would be easy for the algorithm to infer the value of the label due to the very high similarity between datapoints in the training and in the testing sets. By splitting the dataset contiguously we force the algorithm to consider testing data never seen in training, even with the moving window processing.

We added a new figure in the Supplementary (new Figure S2) to show the results in both training and testing.

- Figure 4: adding colorbars could help have an idea of the contour contrast.

We added a colorbar to the figure. We also added colorbars to the (similar) Figures S5 and S9 of the Supplementary.